# Nutrient Estimation from 24-Hour Food Recalls Using Machine Learning and Database Mapping: A Case Study with Lactose

**DOI:** 10.3390/nu11123045

**Published:** 2019-12-13

**Authors:** Elizabeth L. Chin, Gabriel Simmons, Yasmine Y. Bouzid, Annie Kan, Dustin J. Burnett, Ilias Tagkopoulos, Danielle G. Lemay

**Affiliations:** 1Western Human Nutrition Research Center, USDA ARS, Davis, CA 95616, USA; elizabeth.chin@usda.gov (E.L.C.); yybouzid@ucdavis.edu (Y.Y.B.); ankan@ucdavis.edu (A.K.); djburnett@ucdavis.edu (D.J.B.); 2Genome Center, University of California Davis, Davis, CA 95616, USA; itagkopoulos@ucdavis.edu; 3Department of Mechanical Engineering, University of California Davis, Davis, CA 95616, USA; gsimmons@ucdavis.edu; 4Department of Nutrition, University of California Davis, Davis, CA 95616, USA; 5Department of Computer Science, University of California Davis, Davis, CA 95616, USA

**Keywords:** dietary recall, nutrient database, machine learning, database matching

## Abstract

The Automated Self-Administered 24-Hour Dietary Assessment Tool (ASA24) is a free dietary recall system that outputs fewer nutrients than the Nutrition Data System for Research (NDSR). NDSR uses the Nutrition Coordinating Center (NCC) Food and Nutrient Database, both of which require a license. Manual lookup of ASA24 foods into NDSR is time-consuming but currently the only way to acquire NCC-exclusive nutrients. Using lactose as an example, we evaluated machine learning and database matching methods to estimate this NCC-exclusive nutrient from ASA24 reports. ASA24-reported foods were manually looked up into NDSR to obtain lactose estimates and split into training (*n* = 378) and test (*n* = 189) datasets. Nine machine learning models were developed to predict lactose from the nutrients common between ASA24 and the NCC database. Database matching algorithms were developed to match NCC foods to an ASA24 food using only nutrients (“Nutrient-Only”) or the nutrient and food descriptions (“Nutrient + Text”). For both methods, the lactose values were compared to the manual curation. Among machine learning models, the XGB-Regressor model performed best on held-out test data (*R*^2^ = 0.33). For the database matching method, Nutrient + Text matching yielded the best lactose estimates (*R*^2^ = 0.76), a vast improvement over the status quo of no estimate. These results suggest that computational methods can successfully estimate an NCC-exclusive nutrient for foods reported in ASA24.

## 1. Introduction

A 24-hour dietary recall is commonly used to assess dietary intake in the previous 24-hour period. Nutrient intake can be estimated when the recall program is coupled with a nutrient database, but not all recall programs use the same database. Two commonly used recall systems in the U.S. are the National Cancer Institute’s Automated Self-Administered 24-Hour (ASA24®) Dietary Assessment Tool [1,2] and the University of Minnesota’s Nutrition Coordinating Center (NCC) Nutrition Data System for Research (NDSR) [3]. ASA24 is a free, online program where subjects self-report intake [2]. In contrast, NDSR requires a license and data is collected from subjects via an interview by trained personnel. 

The ASA24 system uses several databases, including the Automated Multiple-Pass Method (AMPM), Food and Nutrient Database for Dietary Studies (FNDDS), Food Pattern Equivalents Database (FPED), and the National Health and Nutrition Examination Survey (NHANES) Dietary Supplement Database [4]. NDSR uses the proprietary NCC Food and Nutrient Database. Both databases have comparable nutrient completeness, with FNDDS at 100% and NCC at 92–100% completeness, but the databases differ in the number of nutrients reported. The ASA24 output includes 65 nutrients [5] and licensed 2018 NCC Database files include 166 nutrients and food components. Sixty-two nutrients are shared between the ASA24 output and NCC database. The NCC database also outputs nutrients and food components such as lactose, soluble and insoluble fiber, sugar alcohols, and individual amino acids while ASA24 does not [6]. While both ASA24 and NDSR/NCC database have widespread use and contain thousands of foods, there is no unique identifier to match each food on a one-to-one basis. Only a small number of foods in ASA24 have an exact known counterpart in NCC. Manual lookup of foods reported in ASA24 into NDSR based on text descriptions and nutrient profiles is time-consuming but is currently the only method to obtain values of NDSR-exclusive nutrients. This presents a major hurdle for investigating nutrient intake when using ASA24 if the research question requires assessment of a nutrient absent in the underlying database. 

Our research group is investigating a series of questions that require assessment of a nutrient that is not reported in the ASA24 output: lactose. Most adults worldwide are unable to digest lactose, the primary carbohydrate in milk. Some populations, however, are able to digest lactose into adulthood in a heritable trait known as lactase persistence (LP) [7]. LP genotypes may influence dairy and more specifically, lactose, consumption. Those who are genetically lactase non-persistent (LNP) may become lactose intolerant and experience gastrointestinal discomfort following lactose consumption and therefore may consume different types of or less dairy than those who are LP. Consumption of lactose by LNP individuals may be fermented and alter the gut microbiome, which may have other impacts on health [8,9]. Since lactose is unavailable in the ASA24 output, studying the relationship between diet and LP genetics is limited to using dairy intake as a whole as a proxy for lactose consumption. However, the amount of lactose differs depending on the type of dairy product. Total dairy intake does not provide a good estimate of lactose intake (Figure 1). Fermented products like yogurt and cheese have lower lactose amounts than milk and using the amount (e.g., servings) of dairy consumed would overestimate the amount of lactose consumed. The amount of lactose may even vary between different types of products. Dry, grated parmesan cheese (NCC FoodID 3272), for example, has 0 g of lactose but processed American cheese (NCC FoodID 3204) has 5.41 g of lactose per 100 g of cheese according to the NCC database. Both, however, contribute to the total dairy servings in the ASA24 output. Therefore, estimation of the amounts of lactose for each ASA24 food item would improve the accuracy of these analyses.

Integrating heterogeneous data originating from multiple sources is a challenge in many fields. For example, in healthcare, patient medical information, insurance claims, and vital records data may exist in different databases with no unique identifier available as a crosswalk among the datasets. Record linking methods use common variables shared across datasets to create links for each record. Various types of record linking have been used to merge patient data with high levels of accuracy, although still required some level of manual review [12,13,14,15,16,17,18].

Another method to extract information from a given data set is to use machine learning. In general, machine learning includes models or algorithms that can be used for classification, prediction, dimensionality reduction, and pattern recognition. Predictive machine learning methods use data with a known response variable to first train a model; the model is then used to predict the response variable for new data. Some recent work has been done to leverage machine learning and computer science methods, such as natural language processing, to automate matching food consumption with food composition. FoodEx2 is a standardized food classification and description system developed by the European Food Safety Authority [19]. Since FoodEx2 is currently used to manually link food consumption and composition [20], Eftimov et al. [21] developed a semi-automated method for classifying and describing foods using FoodEx2 by first categorizing food into FoodEx2 food categories using an ensemble of support vector machine, random forests, boosting, and max entropy models followed by Parts of Speech tagging to describe the food, with the highest probability match returned. A combination of the categorization and language processing was then used to further refine the food category. Another study used a computer-assisted, semi-automated method for linking food consumption data collected from different countries and described using different systems with food composition data described using LanguaL [22]. LanguaL, which stands for “Langua aLimentaria” (“language of food”), is a faceted classification system that uses standardized terms to describe food, and has been used to index foods in datasets/databases originating from Europe, the U.S., New Zealand, Canada, and others, although FNDDS and the NCC database have yet to be indexed [23]. Fuzzy matching of food descriptions has also been used to match food consumption with composition tables for data recorded in different languages as well as using English-translated data [24]. When a priori knowledge of the energy content was known, the fuzzy match score and energy content were used as input for a classification tree that predicted the probability of a match being correct. These works demonstrate the utility of computational methods for automating mapping of food consumption to composition databases. However, to our knowledge, no work has been done to extensively map between food composition databases to extend the nutrient information of foods in each database. This would therefore improve the research potential of foods reported through different dietary collection systems. 

The objective of this study was to develop computer-assisted methods to obtain values for NCC-exclusive nutrients for food items reported in an ASA24 output for two scenarios: (1) When researchers have a license to NDSR but not NCC, and (2) when researchers have a license to NCC database files. We use lactose as a case study and estimate values from foods recorded from ASA24 recalls by a California cohort of healthy adults [25]. To address the first scenario, we trained and tested nine machine learning models on the ability to predict the amount of lactose in a given ASA24 food, comparing the predicted lactose values to the lactose obtained by manually looking up ASA24 foods into NDSR. For the second scenario, we developed a database matching algorithm that used only the 62 nutrients shared between the ASA24 output and the NCC database to return the best NCC food match for a given ASA24 food, as well as another algorithm that used both the nutrients and text descriptions of the foods.

## 2. Materials and Methods

### 2.1. Data Sources

#### 2.1.1. 24-hour Dietary Recall Data Collection

The Automated Self-Administered 24-hour recall (ASA24®) system was used to collect dietary recalls from participants in the Nutritional Phenotyping Study, a cross-sectional observational study conducted by the United States Department of Agriculture, Agricultural Research Service, Western Human Nutrition Research Center located in Davis, California, USA (clinicaltrials.gov: NCT02367287) [25]. Foods used for the training and test data sets were retrieved from the first *n* = 214 participants’ recalls. Of these participants, *n* = 63 used the ASA24-2014 version and *n* = 151 used the ASA24-2016 version, which use the Food and Nutrient Database for Dietary Studies (FNDDS) 4.1 (2007/2008) and FNDDS 2011-2012, respectively. The ASA24 FoodCode for each reported food was obtained from the *INFMYPHEI* (ASA24-2014) and *Items* (ASA24-2016) files. The resulting list of FoodCodes was uniquified. The variable corresponding to total dairy servings (D_TOTAL) was used to further select foods to use for this study. D_TOTAL is the sum of the cup equivalents of total milk (D_MILK, including both cow’s and soymilk), total cheese (D_CHEESE), total yogurt (D_YOGURT), and whey for a food. Therefore, D_TOTAL would be useful for selecting foods that were both single-dairy foods (e.g., 2% milk) and foods with dairy from multiple sources (e.g., Macaroni with cheese). One cup equivalent of milk, cheese, and yogurt is defined by the Food Patterns Equivalent Database (FPED) [26] and corresponds to 245 g for milk and yogurt and 1–4.5 oz (28.3–127.6 g) of cheese, depending on the cheese type; whey is not included in the FPED as a separate variable and does not have a weight associated with one cup equivalent. Foods containing an average of at least 0.1 cup equivalent of dairy (D_TOTAL > 0.1) were included to yield 309 dairy-containing foods to be manually entered in NDSR to look up the lactose content. The purpose of this cutoff (D_TOTAL > 0.1) was to limit the number of foods for manual lookup to those that were consumed in relatively large quantities. An additional fifteen dairy-containing foods that were commonly consumed but had an average of less than 0.1 cup equivalents of dairy were added to the full data set, as well as 63 non-dairy (lactose-free) foods (i.e., D_TOTAL = 0), corresponding to a total of 387 unique FoodCodes. Some of the methods described below use text-mining techniques (Section 2.3.2) and we sought to include foods with challenging food descriptions. Therefore, the non-dairy foods included foods with ASA24 Food Descriptions with (e.g., “Milk, almond”) and without (“Lettuce, raw”) dairy-associated words.

Due to some subjects using different versions of ASA24, some of these FoodCodes were only reported in one of the ASA24 versions. Many FoodCodes had the same food description for both versions, although the values of some nutrients (per 100 g of food) may have varied between the ASA24/FNDDS versions. Therefore, if a FoodCode was reported in both ASA24 versions, the food was included in the dataset twice (one for each year) since the nutrient values differed. This corresponded to a grand total of 567 foods output from both ASA24-2014 and ASA24-2016 (with 387 unique FoodCodes) in the complete dataset. Of the 387 FoodCodes, 180 were retrieved from both ASA24-2014 and ASA24-2016, 47 were retrieved from only ASA24-2014, and 160 were retrieved from only ASA24-2016. 

#### 2.1.2. Manual Lookup of ASA24-Reported Foods into ASA24-2016

To estimate lactose values from the ASA24-reported foods, a team supervised by a registered dietitian manually looked up these foods in NDSR (Section 2.1.3). However, before foods could be looked up in NDSR, more information about each food needed to be gathered from the ASA24 system because the manual lookup process relied on answering the NDSR prompts in a similar way as what was used in the ASA24 system. Foods were first entered in ASA24-2016 (ASA24-2014 was retired at time of lookup) to observe dietary recall prompts and answers to use as cues for manual lookup into NDSR 2017 (Figure 2). The 387 unique FoodCodes served as the ASA24 query foods for the manual lookup process. Foods were looked up into ASA24 in batches of 20–30 foods. Each ASA24 query food had a FoodCode, Food Name, and Food Description associated with it. The ASA24 Food Description was used as a first quality control pass to assess the manual lookup process. For clarity, we will use the term “ASA24 Food Name” for a subject’s selection in the ASA24 interface, which can be found in their *MS* (ASA24-2014) or *Responses* (ASA24-2016) file; “ASA24 Food Description” corresponds to the food item described in the ASA24 output analytic files, *INFMYPHEI* (ASA24-2014) or *Items* (ASA24-2016) (Appendix A). The ASA24 Food Name is an undetailed term that subjects use to first search for foods in the ASA24 system; details that are gathered by subjects’ subsequent answers to ASA24 system prompts are used to generate ASA24 Food Descriptions, which are more detailed than the ASA24 Food Name. Therefore, the ASA24 Food Name that a subject selected on the ASA24 interface does not always match the ASA24 Food Description in the output analytic files (Appendix A). For each ASA24 query food, the Food Name was input into the ASA24-2016 system. The resulting ASA24 Food Description was compared to that of the original query. Food Descriptions that matched indicated that the ASA24-2016 system prompts were answered correctly by the manual lookup team. It was necessary to obtain the answers to ASA24-2016 system prompts so that similar answers could be input into NDSR for similar prompts.

#### 2.1.3. Manual Lookup of ASA24-Reported Foods into NDSR

The ASA24 query foods were entered in NDSR 2017 in record-assisted recall format. The ASA24 Food Name was input into NDSR, and the resulting prompts were answered using the ASA24 prompts/answers to guide response selection in NDSR (Figure 1 and Appendix A). For single dairy foods it was usually not difficult to find a match in NDSR because the ASA24 and NDSR food descriptions were similar (e.g., “Yogurt, plain, nonfat milk”; Appendix A). However, it was a challenge to map branded fast foods, mixed dishes, foods where details were “not further specified” (NFS), and foods with multiple options listed in a single Food Description in NDSR. Many ASA24 food descriptions are generic and without brand names even when the subject selected a specific brand in the ASA24 system (Appendix A). To match as closely as possible, the manual lookup team checked the *MS* and *Responses* files for ASA24-2014 and 2016, respectively, to find what brand the subject was reporting. In NDSR, the specific food from the Branded Foods list was entered and reported in output files such as “Taco Bell burrito”, “Burger King burger”, or “Wendy’s burger”. 

Some ASA24 Food Descriptions list several food options in the description for one food reported (e.g., "egg omelet or scrambled egg” (ASA24 FoodCode 32130010), “taco or tostada” (ASA24 FoodCode 58101720), “burrito, taco, or quesadilla” (ASA24 FoodCode 58100010)). NDSR had options for each of the foods listed, but no equivalent option containing both foods (e.g., NDSR has “omelet” and “scrambled eggs” listed as separate foods, but nothing comparable to “Egg omelet or scrambled egg”, Appendix A). In some cases, the foods had differing amounts of lactose according to NDSR/NCC, despite having the same FoodCode in ASA24/FNDDS. For example, scrambled eggs have milk in NDSR while omelet does not. Therefore, to avoid randomly choosing either scrambled egg or omelet in NDSR, the FNDDS-SR links recipe for the given FoodCode was used to guide creation of a User Recipe in NDSR (Figure 2 and Appendix A). A similar process was used for other ASA24 foods that did not have clear NDSR counterparts, such as ASA24 Food Descriptions containing “NFS” (“Not further specified”) (Appendix A). If the FNDDS-SR links recipe contained a food mixture (e.g., recipe or a single Food Description containing multiple foods) as one of the ingredients, a User Recipe was created for that ingredient food mixture in NDSR, resulting in some User Recipes with embedded recipes for the single ingredient (Appendix A). User Recipes were also created to embed raw meat as an ingredient when the FNDDS-SR links recipe contained raw meat as an ingredient. Some recipes in FNDDS listed certain ingredients as raw, with the assumption that those ingredients would eventually be cooked, but it was difficult to find how FNDDS created those cooking assumptions (e.g., moisture loss, nutrient loss, or fat loss). Whereas with NDSR, the cooking assumptions are documented during the food entry process (e.g., weight before or after cooking) and reveal the adjustments in the nutrient reports. 

For all manual lookup foods, including those from User Recipes, the similarity of each NDSR match to the ASA24 query food was rated “high”, “medium”, or “low” confidence by the manual lookup team based on two categories. The first category, nutrient confidence, was based on how similar the values for seven nutrients were (Appendix A). Total energy (kcal) and the following macronutrients: total carbohydrates, total fat, total protein, and sodium, were selected for a simple comparison, and calcium and phosphorous were selected because these nutrients are abundant in dairy-containing foods. The absolute difference between the ASA24 and NDSR values for these seven nutrients were used to assign “high”, “medium”, and “low” ratings for each nutrient using the cut-off values listed in Appendix A, where a “high” rating has a small difference between values and a “low” confidence rating has a large difference in values. The second category, description confidence, was based on how closely the ASA24 and NDSR food descriptions matched. “High”, “medium”, and “low” ratings were assigned for all foods by the same manual lookup team based on their assessment of how similar the prompts were between the two systems (which would influence whether specific answers/food details could be inputted) and the actual words used to describe each food (Appendix A). The overall match confidence level was assigned as “high” if the description confidence rating was “high” and at least five individual nutrients were rated as “high.” A grand total of 491 foods retrieved from ASA24-2014 and ASA24-2016 outputs (corresponding to 327 unique FoodCodes) had an overall “high” confidence rating and included mixed dishes/recipes (e.g., ASA24 FoodCode 58106725, “Pizza with meat and vegetables, regular crust”) as well as single-item foods (e.g., ASA24 FoodCode 14104100, “Cheese, Cheddar”, Appendix A). These confidence ratings were used to assess whether computer-automated mapping performance varied with human confidence.

#### 2.1.4. Creation of Training and Test Data

To train and test computer-automated mapping approaches, the manually mapped data were split into training and test data sets. A total of 567 foods (227 from ASA24-2014 and 340 from ASA24-2016) were used for creation of the training and test data. The data were split into lactose-containing and non-containing foods and then randomly chosen from within those subsets to ensure that the training and test data sets had equal representation of lactose-containing foods. The training data consisted of a grand total of 378 foods, 87 of which had 0 g of lactose according to the manual lookup into NDSR. The test data consisted of a grand total of 189 foods, 44 of which had 0 g of lactose. Therefore, the test data were approximately 33% of the entire data set and both training and test data sets consisted of approximately 23% of non-lactose foods (0 g of lactose). 

### 2.2. Machine Learning Models

All code for the machine learning models can be found at https://github.com/g-simmons/dairyML. Machine learning models were developed using Python 3.6. Custom machine learning model implementations are stored in the GitHub file dairyML/src/dairyml.py. Final versions of each model were exported as binary objects using *pickle*, with separate files for each model. Model hyperparameters were set to the default values unless otherwise stated. All model tuning and training, as well as testing on the held-out data, was performed using Python 3.6. As an extra level of precaution, models were trained and tested by two different team members to ensure that models were never accidentally trained using held-out test data. 

#### 2.2.1. Data Preprocessing

The training data consisted of a total of 378 foods and were used for model development. The test data were held out and were only used for model evaluation. Both the training dataset features were standardized to zero mean and unit variance using *scikit-learn’s StandardScaler* and the test data were preprocessed using the training data mean and standard deviation. Principal Component Analysis (PCA) and *t*-Distributed Stochastic Neighbor Embedding (*t*-SNE) were first used to visualize the training foods in a two-dimensional feature space. (Appendix A). Outliers were removed from the training data using *scikit-learn’s IsolationForest*, an implementation of the methods described in [27] and [28], with the contamination set to 0.013 and all other parameters set to default. *IsolationForest* creates a collection (“forest”) of decision trees. Each tree repeatedly splits the data using a randomly selected threshold value for a randomly selected feature (nutrient), until each sample in the data is “isolated” to one leaf node of the decision tree. *IsolationForest* operates on the assumption that outliers occur less frequently than normal observations and their feature values differ greatly from the normal observations. Accordingly, the number of random splits required to isolate an outlier will be fewer, on average, than what is required for a normal observation. The average number of splits required to isolate each sample is calculated over the forest of random trees and those data points with the lowest average are identified as outliers. For the training dataset, five foods were removed as outliers using the 63 numeric features (62 nutrients and the ASA24 year) for partitioning (Appendix A). 

#### 2.2.2. Baseline Models

Baseline models were developed to confirm that more complex models such as the least absolute shrinkage and selection operator (LASSO), Ridge, feed forward neural network (FFNN) and gradient boosted (XGB) models performed better than naïve predictions (mean, median, median of non-zero values, and perfect classifier plus mean regressor). For the mean baseline model, the prediction of each food was the mean of the lactose values from the training data. For the median baseline model, the prediction was the median value of the lactose values from all the training data. For the “median of non-zero values” model, the prediction was the median of all the non-zero lactose values. For the perfect classifier plus mean regressor, the baseline model predicted 0 when the true value is 0 or the mean lactose amount from the training data when the true value is non-zero; the classification into 0 or non-zero is perfect as it is defined from the data.

#### 2.2.3. LASSO Models

Least absolute shrinkage and selection operator (LASSO) regression is a type of linear regression that avoids overfitting by placing a penalty on the magnitude of the model coefficients. Traditional linear regression fits a model by finding coefficients that minimize the residual sum of squares (RSS). LASSO fits a model by minimizing the RSS *and* the sum of the absolute value of the coefficients (the L1 penalty). Thus, there exists a tradeoff between the closest fit to the training data, and the smallest model coefficients. The hyperparameter *alpha* is used to determine how much the size of the model coefficients is emphasized in this tradeoff. We identified the optimal value of *alpha* by comparing the model 10-fold cross-validation performance (*R^2^*) for 50 values of *alpha* regularly distributed on a log scale between 10^−3^ and 1. This process of identifying the optimal hyperparameter value by comparing model performance at a number of regularly-spaced values is known as *grid search*, and was performed using *scikit-learn*’s *GridSearchCV*. The value of *alpha* that resulted in the best performance was found to be 0.039. All other LASSO hyperparameters were left at the *scikit-learn* default values.

The resulting LASSO model produced some negative predictions of lactose. To resolve this, the LASSO predictor was modified with a lower bound of 0 on all predictions (i.e., negative predictions were clipped to 0). This is hereafter referred to as the “Bounded-LASSO” model. Grid search, as described above, was again used to tune *alpha*, with the exception that 100 intervals was used instead of 50. The best value of *alpha* for Bounded-LASSO was 0.020. 

The resulting Bounded-LASSO model produced some non-zero predictions of lactose for foods known to have zero lactose. To reduce this type of prediction error, we combined the output of the Bounded-LASSO regressor with a logistic regression classifier that first predicts whether lactose values are zero or non-zero (“Combined LASSO”). The element-wise product of the two model outputs was used as the final prediction: When the classifier predicts that a food is lactose-containing, the LASSO output was used as the prediction value and when the classifier predicts that a food is non-lactose-containing, the prediction value was 0. Grid search was used for hyperparameter tuning. The *alpha* hyperparameter for the LASSO regularization was evaluated at 10 values logarithmically distributed between 10^−3^ and 10^−1^. The *C* hyperparameter was used for logistic regression regularization and was evaluated at 10 values logarithmically distributed between 10^−4^ and 10^−6^, as suggested by *scikit-learn’s LogisticRegressionCV* documentation. All other hyperparameters were left at the default values. The best *alpha* value was 0.022 and the best *C* value was 215,443.47. 

#### 2.2.4. Ridge Models

Ridge regression is a modified type of linear regression similar to LASSO. Ridge regression fits a model by minimizing the RSS and the sum of the squares of the coefficients (the L2 penalty, in contrast to the L1 penalty used in LASSO). As in LASSO, the hyperparameter *alpha* controls how strongly coefficients are restricted. 

Like the LASSO models, “bounded” and “combined” versions of Ridge models were produced. Hyperparameter tuning methods were similar to those used for the LASSO models. Grid search on *alpha* was performed at 10 values logarithmically distributed between 10^−3^ and 10^3^ for the Ridge regression model, 100 values logarithmically distributed between 10^−3^ and 10^3^ for the Bounded-Ridge model, and 10 values logarithmically distributed between 10^−4^ and 10^3^ for the Combined Ridge model. The best *alpha* values, as determined by 10-fold CV and *R*^2^, were 59.64, 15.20, and 27.83 for the Ridge regression, Bounded-Ridge, and Combined Ridge models, respectively. The *C* hyperparameter for the logistic regression classifier was evaluated at 10 values logarithmically distributed between 10^−4^ and 10^−6^. The best *C* value was 215,443.47.

#### 2.2.5. Feed Forward Neural Network (FFNN)

Feed forward neural networks (FFNN) are a learning algorithm loosely inspired by the structure of the brain. Within a FFNN, processing units (nodes) are organized in layers that are sequentially connected (an input layer, followed by zero or more “hidden layers”, and finally an output layer). A FFNN learns to map the input data to the response variable by iteratively adjusting the strengths (weights) of the connections between layers. For each sample in the training data, the error between the model output and the actual response variable value is calculated by a user-specified “loss function.” The model weights are then adjusted to decrease this error. The learning rate hyperparameter determines how much adjustment is applied to the weights at each step. This process is repeated for each sample in the training set, gradually improving the model’s predictive performance by adjusting the weights. The whole training dataset may be passed through the algorithm multiple times, where each pass is called an “epoch.” 

Hyperparameter tuning of number of hidden layers, number of nodes per layer, learning rate, *alpha* (L2 regularization), and number of training epochs was performed using Bayesian Hyperparameter Optimization (BHO) via the *hyperopt* package. BHO requires defining an initial probability distribution for each hyperparameter, which were as follows: hidden layers: Uniform linear distribution between 1 and 2; nodes per layer: uniform log between 3 and 100; learning rate: uniform log between 0.001 and 0.1; *alpha*: uniform log between 0.0001 and 0.1; training epochs: normal distribution with mean = 150 and variance = 25. The optimizer sampled combinations of hyperparameters from these distributions and trained and evaluated the resulting model using 5-fold CV with R^2^ as the performance metric. The final model was evaluated using 10-fold CV. After 93 trials, the final hyperparameters were as follows: number of hidden layers = 2; nodes per layer = 50, learning rate = ~0.0012; alpha = 0.029; and training epochs = 185.

#### 2.2.6. Gradient Boosted Trees

A gradient-boosted tree model was implemented using the eXtreme Gradient Boosting (XGB) implementation in the *xgboost* package. The model is an ensemble of trees where new trees are added iteratively to improve the predictions of the existing ensemble. Two XGB models were produced: (1) a regressor-only model where the lactose is predicted from the 62 nutrient variables and the ASA24 year using the gradient boosted tree model (XGB-Regressor) and (2) a model combining a classifier that first predicts whether a food is lactose-free or lactose-containing; a food classified to be lactose-free will have a lactose prediction of 0 and a food classified to be lactose-containing will use the output of the XGB-Regressor model for the lactose prediction (Combined XGB). 

For the XGB-Regressor model, the maximum depth of each tree was set to 9, and the number of columns used by each tree was set to 0.9 based on heuristic evaluation of model performance during cross-validation. The default (squared error) loss function was used. The number of estimators (trees) was tuned using grid search (10-fold cross-validation with 5 repeats, R^2^ selection metric), producing an optimal value of number of estimators = 100. 

To find the optimal maximum depth for the classifier component, the maximum tree depth was first tuned for a classifier-only XGB model by evaluating depth at 7, 9, and 11. The final model hyperparameters for the Combined XGB model were as follows: number of estimators: number of estimators = 100; number of columns per tree for the regressor = 0.9; number of columns per tree for the classifier = 1.0; max depth of the regressor = 9; and max depth of the classifier = 7. Based on these results, for the XGB combined model, the max depth of the regressor component was set to 9 and the max depth of the classifier component was set to 7.

#### 2.2.7. Machine Learning Model Evaluation

The final machine learning models were fit using the training data set with outliers removed (n = 373 foods), exported using *pickle,* and evaluated using the held-out test data of all confidence levels (*n* = 189) as well as only “high confidence” test data (*n* = 152). For all models, the predicted lactose values output by the machine learning models were compared to the lactose from the manual lookup. The performance was evaluated using the following metrics: the coefficient of determination (R^2^), Spearman’s rank coefficient (SRC), Pearson’s correlation coefficient (PCC), and mean absolute error (MAE). Classifier accuracy was also evaluated for the Combined LASSO, Combined Ridge, and Combined XGB models. High *R*^2^, SRC, PCC, and classifier accuracy values, and low MAE are desirable. Results for the training data represent the average values obtained from 10-fold CV. 

### 2.3. Database Matching

All code for the database matching can be found at https://github.com/g-simmons/dairy_matching. Database matching methods were developed using Python 3.6.

Database matching algorithms were developed to return the five most “similar” NCC foods for a given ASA24 query food; the lactose values could subsequently be retrieved. “Similarity” between the ASA24 input food and returned NCC 2018 foods was defined based on the type of matching algorithm. Two types of matching algorithms were developed. The first, “Nutrient-Only” database matching, used only the 62 nutrients in common between FNDDS and NCC. The second, “Nutrient + Text”, used both nutrient information and text information from the food descriptions represented using term frequency- inverse document frequency (TF-IDF). For both Nutrient-Only and Nutrient + Text matching, the initial algorithms were developed using only the training data, but the results from only the training data, only the test data, as well as the training + test data combined were all evaluated.

#### 2.3.1. Nutrient-Only Database Matching

Using the 62 nutrients in common between the ASA24 output and NCC, pairwise Pearson’s correlation coefficients (PCC) were calculated between all entries for the training ASA24 foods and all the foods in the NCC database, resulting in a similarity matrix with the number of rows equal to the number of ASA24 food entries and the number of columns equal to the number of NCC database foods. The input nutrient values were also weighted using the coefficients from the Bounded-LASSO and Bounded-Ridge models as they were the machine learning models with the best performance (the coefficients of the Bounded and Combined models are the same). For Bounded-LASSO- and Bounded-Ridge-weighted Nutrient-Only matching (hereafter referred to as “LASSO-weighted” or “Ridge-weighted”), the absolute values of the coefficients from the Bounded-LASSO and Bounded-Ridge models were used to weight the pairwise similarity calculations. Five NCC food matches were identified by selecting the five foods with the highest PCC value, with the “best match” having the highest value. 

#### 2.3.2. Nutrient + Text Database Matching

Since the Nutrient-Only matching did not use text information (food descriptions), we also evaluated a Nutrient + Text database matching algorithm, which used PCC nutrient similarity combined with TF-IDF representations of the ASA24 output food descriptions and the NCC “short descriptions” for the foods in the NCC database. TF-IDF generates a unique numerical vector to represent the text content of each label. The calculation of the TF-IDF vectors was based on the procedure described in [29]. Manual review of the food descriptions revealed that the first phrase of the description contained the most information, with details following in comma-separated chunks. Therefore, the first comma-separated chunk was repeated to reflect its greater relative importance. Tokenization was then performed by splitting the food descriptions into substrings of 3 characters (“3-grams”, e.g., the 3-grams of “burger” would be “bur”, “urg”, “rge”, and “ger”). The TF-IDF vectors were calculated from the tokenized descriptions using *scikit-learn’s TfidfVectorizer*. 

A similarity matrix of the TF-IDF vectors for the ASA24 and NCC labels was computed using cosine similarity. The unweighted, LASSO- weighted, and Ridge-weighted PCC similarity matrices were calculated as described above. The combined similarity matrix was calculated using element-wise multiplication of the PCC similarity matrix with the TF-IDF matrix. The five food matches with the highest values of row-wise maxima in the combined similarity matrix were returned, with the “best match” having the highest value. 

#### 2.3.3. Database Matching Evaluation

The database matching methods were evaluated using R v.3.6.0. The database matching algorithms were evaluated using the training (*n* = 378 foods), test (*n* = 189 foods), and combined training + test datasets (*n* = 567 foods) of all confidence levels, as well as only the “high confidence” foods (training *n* = 339, test *n* = 152, and combined training + test *n* = 491). The primary evaluation method was comparing the lactose content of the best (i.e., first) match returned by each algorithm to that of the manual lookup match. The *R*^2^, PCC, and MAE values were used as evaluation metrics and the food descriptions of the first matches were also manually reviewed. 

Approximately 23% of each data set contained lactose-free foods (g lactose = 0). We therefore calculated the accuracy of the database matching methods to correctly return lactose-free foods as the first match. We also evaluated the database matching methods independent of the manual lookup output. The median coefficient of variation for the grams of lactose for the top five matches was also calculated. The assumption was that similar foods would have similar amounts of lactose, so a small variation suggests similar foods among the top five matches. A small portion of foods in the NCC database have known FNDDS counterparts or “FoodLinks.” Since the ASA24 FoodCode is derived from the FNDDS FoodCode, we also evaluated how many correct FoodLinks were returned by the matching algorithms. However, this method was limited in that there were only 33 and nine foods in the training and test data, respectively, with known FoodLinks. 

## 3. Results

The median amounts of lactose in the training and test data were 0.484 g and 0.772 g, respectively. The maximum amounts were 21.31 g and 13.27 g for the training and test data, respectively. The distributions of lactose for the test and training set were not significantly different (Kolmogorov–Smirnov test, *p* > 0.05, data not shown). 

### 3.1. Machine Learning Models

The training data were used to develop machine learning models to predict the amount of lactose for a new food and the test data were used to evaluate model performance (see Section 2.2). Baseline models using naïve predictors were first created as comparison tools for the LASSO, Ridge, FFNN, and XGB models. Comparison of the machine learning model performance metrics indicated that all the model predictions were better than naïve guessing (Appendix A). 

#### 3.1.1. Training Results

For the training validation results, the LASSO and Ridge models had similar performance metrics (Table 1). For both, the bounded models outperformed their “un-bounded” counterparts. A classifier that predicted whether a food had zero lactose or not was added to the Bounded-LASSO and Bounded-Ridge models. The same classifier was used for both the Combined LASSO and Combined Ridge models and had an accuracy of 0.89. The classifier improved the *R*^2^ values for both (*R*^2^ = 0.64 for both Combined LASSO and Ridge) and reduced the error (Combined LASSO MAE = 0.86 and Combined Ridge MAE = 0.87). The FFNN model outperformed all the LASSO and Ridge models, with an *R*^2^ of 0.74 (Table 1). The XGB models had the overall best performance during training validation, with the highest R^2^ and lowest MAE values of all the models. The classifier accuracy of the Combined XGB model (accuracy = 0.97) was also higher than the LASSO and Ridge classifier (accuracy = 0.89 for both).

#### 3.1.2. Test results

When testing the models using all of the independent held out test data (*n* = 189), the XGB-Regressor model had the highest *R*^2^ and PCC and the Combined XGB model had the highest SRC and lowest MAE (Table 1 and Figure 3a), although the overall performance was similar between the two models. The LASSO (*R*^2^ = 0.32), Bounded-LASSO (*R*^2^ = 0.30), and Combined LASSO (*R*^2^ = 0.32) models had similar *R*^2^ values as well. The Ridge models had the poorest performance, having the lowest R^2^ values. 

With the exception of the Combined XGB model, all models predicted “Salmon, raw” (ASA24 FoodCode 26137100) to have a non-zero lactose value. Notably, all three LASSO and Ridge models predicted high amounts of lactose, and removal of this food from the test data improved model performance (Table 1, Appendix A). The FFNN model predicted negative lactose values, but removal of this food did not substantially improve model performance (Table 1). When “Salmon, raw” was removed from the test data, the Combined LASSO model had the highest R^2^ value (Table 1).

We also tested a subset of foods from the test data where the NDSR manual lookup match was considered “high confidence.” It was difficult to find obvious matches for some query foods, especially for mixed dishes or recipes; the confidence ratings were a useful way to gauge the quality of a manual lookup and how reliable the lactose value may be for a given food. Of the 189 test foods, 152 were considered “high confidence.” Model performances for the “high confidence” test foods were similar to the performance when using all the test foods (Appendix A). Model performance using “high confidence” test foods were better (higher R^2^, PCC, SRC, and classifier accuracy, and lower MAE) than when using all the test foods (Appendix A). The performance of the of the LASSO, Bounded-LASSO, and Combined LASSO models were similar to that of their Ridge counterparts when using only the “high confidence” test foods. The XGB-Regressor and the Combined XGB models had the highest R^2^ values (R^2^ = 0.53) and lowest MAE (MAE = 0.96 and 0.92, respectively).

#### 3.1.3. Feature Importances

We examined the feature importances (coefficients) for the best-performing version for each model type (LASSO, Ridge, FFNN, or XGB). The coefficients of the Bounded models were the same as that of the Combined models because the Combined models use the Bounded regressor. The magnitude of the coefficient for each feature indicates how much each feature contributes to the overall lactose estimate. The coefficients of the Bounded-LASSO and Bounded-Ridge models were similar (Appendix A). Of the 63 features (62 nutrients and ASA24 year), 29 had coefficients of zero for the Bounded-LASSO model. Sugar and iron were in the top ten (based on the absolute magnitude of the coefficient/weight) feature importances for all models (Figure 4). Sugar was a positive predictor of lactose for all models and iron was a negative predictor for Bounded-LASSO, Bounded-Ridge, and FFNN (the feature importances for the XGB model do not include directionality information). Additionally, potassium was a positive predictor of lactose, and cholesterol and fiber were negative predictors for the Bounded-LASSO, Bounded-Ridge, and FFNN models. However, these three nutrients were not among the top ten feature importances for the Combined XGB model. Total kcal as well as the total primary energy sources, total fat, total protein, and total carbohydrate, were important features for the Combined XGB but not the others. The ASA24 year was not in the top ten feature importances for any of the models, with a near-zero feature importance value for the bounded-LASSO, bounded-Ridge, FFNN, and XGB models (−0.015, −0.049, −0.003, and 0.001, respectively, Appendix A).

### 3.2. Database Matching

The training data were used for initial development of the database matching. Though no parameters were specifically tuned, and the database matching method does not “learn” anything from a given dataset, the length of the TF-IDF vector is equal to the number of unique 3-grams in the corpus (dataset). A bigger input corpus typically contains more unique 3-grams and thus the TF-IDF vectors will vary depending on the corpus size. Therefore, the results of the Nutrient + Text similarity matching depends on the size of the input dataset. For the Nutrient-Only matching, the results are independent of the input data size. We used only the training data (*n* = 378), only the test data (*n* = 189), and a combined training + test dataset (*n* = 567) to evaluate matching using different input sizes, as well as for comparability to the machine learning results. We evaluated all the foods in the training, test, and combined datasets as well as the “high confidence” foods. 

R^2^ and Mean Absolute Error (MAE) were used to evaluate the performance of the database matching methods. The NCC food that was the first match (i.e., highest similarity) was used to evaluate the database matching. The match generated from the manual lookup process was used for comparison. For each ASA24 query food, the lactose for the NCC first match and the manual look up were compared. The ASA24 query foods were looked up into NDSR, which does not provide the NCC FoodID. Therefore, identification of exact matches between the matching algorithms and the manual lookup process could not be easily conducted. 

#### 3.2.1. Nutrient-Only Database Matching

The Nutrient-Only matching algorithm matched ASA24-reported foods to foods in the NCC database using the nutrients shared between the ASA24 output and NCC database. Unweighted, LASSO-weighted, and Ridge-weighted Nutrient-Only matching algorithms were evaluated (see Section 2.3.1). 

##### Comparison of Lactose to the Manual Lookup

When comparing the lactose value of the first NCC match (i.e., the NCC food with the highest PCC value) to that of the manual lookup match, the *R*^2^ the unweighted and Ridge-weighted Nutrient-Only matching were the same for the training data (*R*^2^ = 0.51) but the Ridge-weighted algorithm had a higher MAE (Table 2). The LASSO-weighted matching had the lowest R^2^ and the highest MAE of all three Nutrient-Only algorithms for the training data (*n* = 378, Table 2). The low *R*^2^ was primarily driven by a single ASA24 food, “Cocoa, whey, and low-calorie sweetener mixture, lowfat milk added” (ASA24 FoodCode: 11516000), which was matched to “milk, unprepared dry powder, lowfat” (NCC FoodID: 4351). Removal of this food improved performance (Table 2, Appendix A). When using only “high confidence” training foods (*n* = 339), the *R*^2^ values were higher for all the algorithms than when using all the training food (regardless of confidence level); the Ridge-weighted Nutrient-Only had the highest *R*^2^ and the LASSO-weighted had the lowest *R*^2^ using high confidence training foods (Appendix A).

For the test data (*n* = 189), the unweighted and Ridge-weighted Nutrient-Only matching algorithms had the highest *R*^2^ (*R*^2^ = 0.32), compared to LASSO-weighted (*R*^2^ = 0.19). The unweighted Nutrient-Only matching also had the lowest MAE (MAE = 1.18, Table 2). All R^2^ values were higher for all the algorithms when using only “high confidence” test foods (*n* = 152) compared to when using all the test foods. The *R*^2^ was highest for the Ridge-weighted model (*R*^2^ = 0.51) but the unweighted Nutrient-Only had the lowest MAE when using the high confidence foods (MAE = 0.97, Appendix A). 

When using all the training and test data combined (*n* = 567), the Ridge-weighted and unweighted Nutrient-Only algorithms had the same *R*^2^ (*R*^2^ = 0.44) but the unweighted algorithm had a lower MAE (unweighted MAE = 0.87 and Ridge-weighted MAE = 0.91, Table 2); LASSO-weighted matching performance was the poorest, even after removal of “Cocoa” (FoodCode 11516000). Both the *R*^2^ and MAE were improved when using only “high confidence” foods for all algorithms, with the Ridge-weighted matching having the highest *R*^2^ (*R*^2^= 0.69) and the unweighted with the lowest MAE (MAE = 0.67, Appendix A). A grand total of 123 foods had the same NCC food as the first match for all three matching algorithms. 

In summary, the unweighted and Ridge-weighted Nutrient-Only algorithms had the highest *R*^2^ values for the training, test, and combined (training + test) datasets. The unweighted Nutrient-Only algorithm had the lowest MAE for all datasets. 

#### 3.2.2. Nutrient + Text Database Matching

The Nutrient + Text database matching used the shared nutrients and text information (food descriptions) to match NCC foods to a given ASA24 food. Unweighted, LASSO-weighted, and Ridge-weighted Nutrient + Text matching algorithms were evaluated (see Section 2.3.2).

##### Comparison of Lactose to the Manual Lookup

The matching algorithms were evaluated using the training data, comparing the first match (i.e., the food with the highest combined similarity matrix) and the manual lookup. Addition of TF-IDF greatly improved the unweighted matching; the *R*^2^ was 0.78 and the MAE was 0.48, compared to 0.51 and 0.71 for the Nutrient-Only matching (Table 2). However, LASSO-weighted and Ridge-weighted Nutrient + Text matching had poor performance on the training data, with *R*^2^ values of 0.20 and 0.23, respectively. For both, the poor *R*^2^ was explained by two foods: ASA24 FoodCodes 11513100 (Cocoa and sugar mixture, whole milk added, from ASA24 2016) and 11514100 (Cocoa, sugar, and dry milk mixture, water added, from ASA24 2016)); removal of these foods increased the *R*^2^ to 0.57 and 0.65 for LASSO-weighted and Ridge-weighted, respectively (Table 2, Appendix A). When using only “high confidence” training foods, the *R*^2^ increased for all the Nutrient + Text algorithms and the unweighted Nutrient + Text matching had the highest *R*^2^ (0.85) and lowest MAE (0.38) (Appendix A). 

For all the test data (*n* = 189), all Nutrient + Text matching algorithms outperformed their Nutrient-Only counterparts (Table 2). The Ridge-weighted matching had the highest *R*^2^ (0.75, Figure 3b), and the unweighted algorithm *R*^2^ was similar (0.72). The LASSO-weighted *R*^2^ was the lowest (0.64, Table 2). When using the “high confidence” test data, the *R*^2^, PCC, and MAE values were similar among all three Nutrient + Text algorithms (Appendix A). For all Nutrient + Text algorithms, the *R*^2^ values for the “high confidence” test data were lower than when using all of the test data. 

When using the training and test data combined (*n* = 567), the unweighted Nutrient + Text matching had the highest *R*^2^ (0.76) and lowest MAE (0.54). Low performance for the LASSO-weighted Nutrient + Text, and Ridge-weighted Nutrient + Text was explained by the same few foods that were driving low performance in the training data. Even after removal of these foods, the unweighted Nutrient + Text still had the highest *R*^2^ (Table 2, Appendix A). There were similar findings using only the “high confidence” foods in both the training and test data (*n* = 491, Appendix A).

In summary, the unweighted Nutrient + Text algorithm had the highest *R*^2^ and lowest MAE for the training dataset and combined dataset (training + test), and the Ridge-weighted algorithm had the highest *R*^2^ and lowest MAE for the test data. 

##### Comparison of Nutrient + Text Outputs Between Datasets

The size of the input data may influence the TF-IDF values. Therefore, the Nutrient + Text values differed between the training (*n* = 378) or test (*n* = 189) data compared to the combined training + test dataset (*n* = 567). As a result, some ASA24 query foods had different first matches returned by the Nutrient + Text algorithms when comparing across datasets (Appendix A). On the other hand, the calculation of the PCC matrix was not influenced by the input data size. Therefore, the outputs of the Nutrient-Only algorithm are not influenced by the size of the training, test, or combined datasets and the same first match is returned for a given ASA24 query food. 

For the unweighted Nutrient + Text matching, a total of four foods had different first matches when comparing results from the training or test to that of the combined training + test data (Appendix A). A total of five foods differed for the LASSO-weighted, and a total of six foods differed for the Ridge-weighted Nutrient + Text matching. In all cases, the first match from the combined training + test dataset was the second match for the training-only or test-only data set. Notably, the ASA24 query food, “Milk, cow’s, fluid, 1% fat” (ASA24 FoodCode: 11112210) was matched to lactose-free milk (“Milk, lactose reduced (Lactaid), 1% fat (lowfat)”) as the first match when using the combined training + test dataset. When using only the test data, it was matched to “Milk, whole” for both the LASSO-weighted and Ridge-weighted Nutrient + Text matching algorithms (Appendix A). Essentially, most of the matches returned for a food in the training and test data were the same when the same food was used in the combined training + test data. 

#### 3.2.3. Comparison of the First Matches Between Database Matching Algorithms

We also manually inspected and compared the NCC first match foods to that of the ASA24 query food to further evaluate the quality of matches. R^2^, PCC, and MAE performance metrics are solely based on the grams of lactose. These metrics do not inform on whether a matched food is actually an appropriate match given the ASA24 query food. Therefore, we also manually reviewed the food descriptions of the food matches. Here, we present a few examples of discrepancies between the first matches from the Nutrient-Only and Nutrient + Text.

For example, for the ASA24 query food, “Milk, almond”, the Nutrient-Only matches are all lactose-free foods but are not actually almond milk. Though the unweighted and Ridge-weighted Nutrient-Only matches are also plant-based alternative “milk” products, the LASSO-weighted Nutrient-Only returned a Nabisco SnackWell product. For many ASA24 input foods, the matches were more similar for the Nutrient + Text algorithms than those of the Nutrient-Only algorithms, corresponding with the improved *R*^2^ and MAE values. Notably, addition of text data correctly distinguished between lactose-reduced and regular milk when the ASA24 query food was lactose-reduced milk (Table 3). 

Many Nutrient + Text matches were also non-branded foods compared to the Nutrient-Only counterparts, regardless of whether weighting was used. For the unbranded ASA24 query food, “Yogurt, plain, whole milk,” the difference between the branded (Nutrient-Only) and non-branded (Nutrient + Text) matches did not influence the amount of lactose. Nutrient + Text algorithms usually returned non-branded foods, but occasionally would return branded food matches (e.g., ASA24 input food description: “High protein bar, candy-like, soy and milk base”). Sometimes a non-branded ASA24 food description may correspond to a branded food. The ASA24 Food Name, “McDonald’s Cheeseburger,” corresponds to the ASA24 Food Description, “Cheeseburger with tomato and/or catsup on bun” (Appendix A). Therefore, the brand name is not apparent from the food description but can be found by inspecting the *MS* or *Responses* file (Appendix A). 

Food Descriptions containing multiple foods were challenging. For example, for “Clam chowder, NS, as to Manhattan or New England style”, the Nutrient + Text algorithms indeed selected one type of clam chowder (Manhattan style), but it was not the type that was actually reported (New England style). On the other hand, the Nutrient-Only matches were not related to clam chowder. 

Overall, the Nutrient + Text database matching method had superior performance to the Nutrient-Only method. 

#### 3.2.4. Variation in Lactose, and Matching to FoodLinks and Lactose-Free Foods

For the Nutrient-Only matching, the unweighted algorithm had the lowest variation for the training (variation = 0.39), test (variation = 0.52), and combined training + test data (variation = 0.44). The Ridge-weighted Nutrient + Text matching had the lowest variation for all three datasets (variation = 0.54 for all datasets). The variation for the unweighted Nutrient + Text matching for the test data were also 0.54. For all datasets, the LASSO-weighted Nutrient-Only and LASSO-weighted Nutrient + Text matching had the highest variation (Table 2). 

There were 33 and nine ASA24 foods in the training and test data, respectively, with known matches (“FoodLinks”) to the NCC database (Appendix A). FoodLinks are therefore true matches between FNDDS/ASA24 FoodCodes and the NCC database. Though only a few foods had FoodLinks, evaluating the FoodLinks is still informative of the accuracy of each matching method. The Nutrient + Text matching algorithm correctly returned more FoodLinks as the first match, as well as within the first five matches, compared to the Nutrient-Only matching (Appendix A). The unweighted Nutrient-Only and unweighted Nutrient + Text matching returned the highest number of FoodLinks as the first match (13 and 19, respectively) and within the first five matches (23 and 30, respectively) for the training data compared to the LASSO- and Ridge-weighted counterparts. The unweighted Nutrient-Only matching had the highest number of FoodLinks returned as the first match in the test data (*n* = 3). For the test data, the same number of FoodLinks were returned for the unweighted, LASSO-weighted, and Ridge-weighted Nutrient + Text matching. 

There were 87 and 44 lactose-free foods in the training and test datasets, respectively (Appendix A). Regarding the training data, the unweighted Nutrient + Text algorithm was the most accurate and returned 70 lactose-free foods (80.5%); the unweighted and Ridge-weighted Nutrient-Only both returned 65 (74.7%) foods with zero lactose. Regarding the test data, the Ridge-weighted Nutrient-Only and Ridge-weighted Nutrient + Text matching, as well as the unweighted Nutrient + Text matching, all returned 38 (86.4%) lactose-free foods. 

In summary, for all three datasets, the unweighted Nutrient-Only matching had the lowest variation in lactose among the top five matches while the Ridge-weighted Nutrient + Text matching had the lowest when compared to the LASSO- and Ridge-weighted counterparts. Overall, the unweighted Nutrient-Only and unweighted Nutrient + Text algorithms correctly returned more FoodLinks and lactose-free foods as the first match compared to the LASSO- and Ridge-weighted counterparts. 

## 4. Discussion

Given that there is no automated way to match foods from the ASA24 output to the NCC database, we explored the use of machine learning models and database matching methods to estimate the amount of lactose, an NCC-exclusive nutrient, from foods output by ASA24 in two scenarios: (1) The user has a license to NDSR, but not NCC, and (2) the user has a license to the full NCC database. 

For the first scenario, machine learning models predicted lactose from the nutrients shared between the two databases, using the values obtained from ASA24 reported foods. The XGB-Regressor model had the highest test R^2^ (0.33) when comparing the lactose prediction with the lactose retrieved from manual lookup into NDSR. For the second scenario, database matching algorithms using either nutrient-only information (Nutrient-Only matching) or nutrient and text information (Nutrient + Text matching) from the food descriptions returned the most similar NCC entry for a given ASA24 food. When compared to the Nutrient-Only matching algorithms, the Nutrient + Text algorithms returned more logical matches and the resulting lactose amounts were closer to that of the NDSR manual lookup match. For all three datasets, the Nutrient + Text matching had higher R^2^ values than the Nutrient-Only matching. Overall, the unweighted Nutrient + Text matching had the highest R^2^ values for the training (0.78) and combined training + test (0.76) datasets, and the Ridge-weighted had the highest for the test data (0.75). When comparing the machine learning and database matching methods using the test data, the lactose values of the Ridge-weighted Nutrient + Text database matching correlated better with that of the manual lookup compared to the XGB-Regressor model (Figure 3).

Linking food databases together can increase the research potential of dietary data. Many of the variables output by ASA24 such as the FoodCodes, food descriptions, and nutrients are derived from FNDDS, which is now part of FoodData Central [30]. FNDDS is associated with other research projects and databases such as What We Eat in America (WWEIA) and the Food Patterns Equivalent Database (FPED) [31]. The same FNDDS FoodCodes are shared by other USDA databases such as the Center for Nutrition Policy and Promotion Food Prices Database, making it easy to study intake in relation to different factors [32]. FoodData Central includes foods from FNDDS 2013–2014 as well as other food databases. Some foods in FoodData Central have lactose values, but this constitutes a very small (<2%) portion of all the food entries in FoodData Central, with only 489 foods containing >0 g of lactose per 100 g of food [33]. In contrast, all foods in the NCC database have a lactose value, with approximately 44% of foods containing more than 0 g of lactose. Therefore, while ASA24 results can be easily linked to other databases since the outputs use FNDDS FoodCodes, the research potential could still be further expanded by linking to the NCC database to estimate additional nutrients like lactose.

A growing body of work has used different computer-assisted techniques to map food consumption data with food composition databases [21,22,24]. These methods mostly rely on correctly matching foods based on the text descriptions using natural language processing techniques. Limited work has been done to link food composition databases to each other. Though LanguaL and ontology-based systems are language-independent and may be used to link foods from different composition databases [20,23], these systems do not utilize nutrient information available for database foods and only semantically link foods. The present study is the first study to use both text descriptions and nutrient profiles to link foods between food composition databases. 

### 4.1. Machine Learning Models

We evaluated machine learning models to predict lactose from the nutrients common between the ASA24 output and the NCC database. All models had lower performance using the held-out test data compared to the results obtained using 10-fold CV of the training data (Table 1). Notably, the regular, Bounded, and Combined LASSO and Ridge model test performances were strongly influenced by the food, “Salmon, raw,” and excluding this food improved the test performances (Table 1). Both Ridge and LASSO regression use regularization to reduce the coefficients of non-predictive features to create a parsimonious model, thereby reducing the tendency to overfit and improving generalizability. However, there was no comparable food to “Salmon, raw” in the training data, suggesting that the poor test performance was due to variation between the training and test dataset features. The XGB models had the biggest discrepancy between the training and test performance, which could not be explained by anomalous foods such as “Salmon, raw.” Similarly, the low performance of the FFNN was not substantially improved after removal of “Salmon, raw,” indicating overfitting to the training data. The tendency towards overfitting increases as the number of learned model parameters (model complexity) goes beyond the number of training examples. L2 regularization was applied to the FFNN model, and regularization was applied to the XGB models by limiting the maximum tree depth and features used to fit each tree (feature dropout), but additional steps may be taken to reduce overfitting. Future work using tree dropout in addition to feature dropout for the XGB models, as well as feature dropout for the FFNN may be considered [34,35]. 

The coefficients of the Bounded-LASSO and Bounded-Ridge models were biologically relevant. Riboflavin (vitamin B2), choline, and potassium are all found in animal products, and dairy products like milk are valuable dietary contributors for these nutrients [36,37,38]; all three were positive predictors of lactose for the Bounded-Ridge and Bounded-LASSO models (Figure 4). Potassium was also a positive predictor of lactose for the FFNN model. Sugar was a positive predictor of lactose for the Bounded-Ridge, Bounded-LASSO, and FFNN models. This would be expected, given that lactose is the primary milk sugar and that the selection of ASA24 query foods was biased towards dairy-containing foods. It was also unsurprising that iron, which is low in dairy products, and fiber, which is absent in dairy, were strong negative predictors of lactose. Cholesterol and lauric acid, both of which are found in animal products, were strong negative predictors of lactose for the Ridge and LASSO models, as well as cholesterol for the FFNN, and may be discriminating dairy-containing foods from other animal products. It is important to note that these findings may be a result of the small number of foods used in the study which were biased towards dairy-inclusive foods. A small number of foods were lactose-free and many of the foods containing small amounts of lactose were mixed dishes and unlikely to accurately represent all the types of mixed dishes in the entire FNDDS database.

### 4.2. Database Matching

Nutrient-Only and Nutrient + Text database matching were used to link foods output by ASA24 to NCC foods. We weighted the nutrient features for the database matching method using the Bounded-LASSO and Bounded-Ridge model coefficients, assuming that shrinking the importance of non-predictive nutrients of lactose would improve the matches. However, the LASSO-weighted Nutrient-Only and Nutrient + Text algorithms had the lowest R^2^ for all datasets (training, test, and training + test foods). After removal of outliers, the performances for the unweighted and Ridge-weighted algorithms were similar (Table 2). Though the LASSO and Ridge coefficients inform of which nutrients are predictive of lactose, it appears as though all nutrients provide some information that is useful for generating a match. Constraining the number of nutrients (as in the case of LASSO) or shrinking the relevance of some nutrients (as in the case of Ridge) therefore did not improve the matching algorithm and shows that database matching using unweighted algorithms can be successful without evaluating machine learning models first. The Nutrient + Text algorithms returned more first matches that were the correct NCC FoodLink. It is important to acknowledge that the NCC and FNDDS databases contain thousands of foods, and that many foods within a single database are similar and differ in only a few details. Therefore, though some ASA24 query foods may not have been matched directly to their corresponding FoodLink, an acceptable match may still have been returned but is not captured in Appendix A. For example, the ASA24 food, “Cheese, mozzarella, NFS” (FoodCode: 14107010) has two NCC FoodLinks: (1) “Cheese, cheese curds, plain” and (2) “Cheese, string, regular. However, the unweighted Nutrient + Text algorithm returned “Cheese, mozzarella cheese, whole milk” as the best match. The LASSO- and Ridge-weighted Nutrient + Text also returned mozzarella cheese but of different fat percentages. These results were similar to the manual lookup match, “Mozzarella cheese, unknown type” (Table 3). Therefore, though the algorithms may not always identify FoodLinks, the best matches were still acceptable and failure to identify FoodLinks does not necessarily indicate poor performance.

We also manually inspected the food descriptions of the best matches returned by the Nutrient-Only and Nutrient + Text algorithms to explain any discrepancies in lactose. For the “outlier” foods that were driving the low performance of some of the models, the large discrepancy in lactose values was a result of hot cocoa-related foods returning unprepared dry mixes (Table 3 and Appendix A). Since the grams of lactose is reported on a per-100 g of food basis, the amount of lactose returned was for 100 g of dry hot cocoa mix, but only a fraction of the product was actually consumed since it was mixed with water. Both ASA24 and the NCC database contain foods/powders that are reconstituted and not reconstituted, including cocoa powders that are not reconstituted; however, these were not included in the training or test datasets because they had 0 servings of dairy and we selected the majority of lactose-containing foods by including only those with ≥0.1 average servings of dairy reported in the study. 

There were some instances where the lactose values were similar between the NDSR manual lookup and the best match food, but was clearly incorrect given the ASA24 query food, as in the case of the Nutrient-Only matches for “Milk, almond, ready-to-drink” (Table 3). One challenge that cannot be resolved using the current methods is that a “correct” match can never be learned from ASA24 foods with multiple foods contained in a single description. For the lactose case study, this is especially problematic when two foods with very different lactose amounts are contained in the same Food Description such as “Clam chowder, NS, as to Manhattan or New England style” (Table 3). The dietetic technicians returned cream-based New England style clam chowder for the manual lookup match, but this information was retrieved by reviewing the subject’s *MS* or *Responses* file. Manual review of individual response files quickly becomes time-prohibitive for more than a few foods. 

Nonetheless, regardless of whether any weighting was used, the Nutrient + Text matches were typically more similar to the manual lookup matches and to the ASA24 query food when compared to the Nutrient-Only matches. This corresponded with lactose values more similar to those of the manual lookup matches (Table 2). Though both Nutrient-Only and Nutrient + Text matching returned some kind of cheese for the ASA24 query food “Cheese, Cheddar,” only Nutrient + Text differentiated the specific variety of cheese (Table 3). Basically, TF-IDF was better able to fine-tune the outputs to match the details of a query food. Importantly, the Nutrient + Text algorithms differentiated between lactose-containing and lactose-free milk (Table 3). The Nutrient-Only algorithms matched whole cow’s milk to lactose-reduced whole milk, which would therefore grossly underestimate the amount of lactose consumed. Additionally, fewer branded foods were returned by the Nutrient + Text matching compared to the Nutrient-Only matching. Some of the ASA24 query foods were actually branded (e.g., McDonald’s Cheeseburger, Appendix A) but the brand was not obvious from the food description and can only be learned by manually reviewing the ASA24 *MS* or *Responses* files. Food Data Central includes the USDA Global Branded Foods Products Database, but it does not use any FoodCodes or other IDs to easily link to the FNDDS counterparts [30]. Ideally, future updates to the ASA24 system would retain brand information for all foods in the output. Since the brand information is currently lost in the ASA24 output food description, language processing techniques such as TF-IDF cannot recover this information. 

### 4.3. Time Required for Manual Lookup and Computer-Assisted Methods

Computer-assisted methods were substantially faster than the manual lookup method. The manual lookup process required that the team first learn how to use both the ASA24 and NDSR systems and understand the underlying databases. Once the team became proficient with the dietary entry tools and databases, lookups required an average of 28 minutes per food. The time required to look up ASA24-reported foods into NDSR depended on the type of food and whether recipes needed to be made. For example, simple, single-ingredient foods (e.g., 2% milk) had fewer input prompts compared to a mixed dish with more details (e.g., Pizza with meat and vegetables, regular crust). On the other hand, once the models and algorithms were finalized, the computer-assisted methods took less than 30 minutes to execute. Outputs for the entire batch of ASA24-reported foods were returned at once, representing a 99.7% reduction in time. Reviewing whether matches were appropriate or whether predicted lactose was logical required a few minutes per food.

### 4.4. Limitations and Future Work

This study is the first to incorporate both nutrient and text (food description) information for merging two food composition databases. These results are promising and demonstrate the potential for machine learning and database matching methods for this field. However, we do acknowledge that this study has some limitations. Notably, the number of foods used for the training and test datasets were small as we were limited by time required for the manual lookup process. The selection process for the foods was biased towards dairy-containing foods and some food groups are surely underrepresented (e.g., fish, seeds, nuts, and meats). This influenced the performance of the machine learning models, as some food groups were represented in the test data but not the training data (Table 1). Therefore, the models had not “learned” how to predict these unseen features, contributing to the poor test performance. The proportion of lactose-free foods was approximately the same for both the test and training data but many of these foods were “core” foods (e.g., olive oil, raw lettuce, and black tea) and few were mixed dishes or recipes. In contrast the lactose-containing foods included a larger proportion of mixed dishes, and this complexity may increase learning difficulty for the machine learning models. Increasing the number and types of foods in the training data may alter the features (nutrients) important for predicting lactose and may also improve generalizability of the machine learning models. A larger dataset would also likely change the Nutrient + Text matching output since the input data size directly influences the calculation of the TF-IDF vectors. 

The foods used in this study were retrieved from ASA24 outputs where subjects reported dietary intake using either ASA24-2014 or ASA24-2016. However, all foods were looked up in ASA24-2016 during the manual lookup process since ASA24-2014 was retired at time of lookup. We do not expect that looking up foods recorded in ASA24-2014 into ASA24-2016 would significantly change any results since the purpose of the lookup was to obtain the recall prompts and answers, but we do acknowledge this discrepancy. Similarly, the ASA24 query foods were looked up into NDSR 2017 which uses the NCC 2017 database, but our database matching algorithms used the NCC 2018 version. Therefore, there may be differences in lactose values and food descriptions between years. Just as the ASA24 output food description is not always identical to that of FNDDS, the food descriptions between NDSR and the NCC database are also not always identical. The output of the matching algorithm is the NCC Short Food Description, but this is not part of the NDSR output so the manual lookup and algorithm matches may use different text to describe the same food. 

The database matching algorithms compared each ASA24 query food to each NCC food. While this is the most comprehensive method since every NCC food is a potential match, it is not the most parsimonious method and is computationally expensive for a larger dataset. Future work using clustering, where similar database records are grouped and then only corresponding clusters are searched against each other for matches, could be used to reduce the number of comparisons [39]. For example, clustering foods by food groups would be an easy way to reduce the computational requirements for matching but may have limited use for mixed dish recipes if they do not belong to the expected cluster. The ASA24 output and NCC flat files may also be pre-processed. Filtering the NCC database to include only un-branded foods so only generic foods can be returned. Using a match that is a generic version of the branded food could be preferable to the time investment needed to review individual response files for studies where brand information is not important. This would also resolve any issues where un-branded ASA24 query foods are matched to branded foods (e.g., Yogurt in Table 3). In addition to removing branded foods, the terms “NS” or “NFS” in the ASA24/FNDDS food descriptions can be changed to “unknown type” to match the language used by NCC. This would improve matching foods with details that are “not further specified.”

We also recognize that these methods have so far only been applied for estimation of lactose. These methods may be adapted for other nutrients but would likely require additional tuning, as all models and algorithms were evaluated using lactose and because our training and test data were biased towards dairy-containing foods. Additionally, the database matching results still required manual review of the outputs, which is subject to human error and bias. Nonetheless, reviewing the best matches from the NCC database was much faster than individually looking up foods into NDSR. Future improvements to the database mapping scheme might consider a rule where foods are flagged for manual review based on, for example, the variation in lactose (or the nutrient of interest) among the top five matches or foods with an ambiguous ASA24 output (e.g., containing more than one food per food description). In such ambiguous cases, there may be no ideal match. 

## 5. Conclusions 

Lactose is just one of many nutrients unavailable from the ASA24 output and these methods demonstrate the feasibility of using computer-assisted approaches to extend the research potential of ASA24 data. For research questions that are specific to a nutrient not output by the ASA24 system, it would be easiest to use the NDSR system. However, this may not be possible due to cost limitations or participant burden, or in the case when ASA24 has already been used. Therefore, using these methods are potential alternatives. Ultimately, database matching using both nutrient and text information is the most accurate and time-effective method but requires a license to the NCC database. For those who have a license to NDSR but not the NCC database, our study suggests that there is potential to use machine learning models to predict the amount of lactose. However, this method requires a large labeled dataset and careful model training.

Though both the machine learning and database matching results required some manual review, the lactose estimated from both methods were similar to those obtained by the manual lookup method and presented a 99.7% decrease in the time required to obtain estimates compared to the alternative of looking up individual food items. Furthermore, considering that two different versions of ASA24 and NCC were used, and that many mappings are ambiguous or may not have a clear match, an R^2^ of 0.76 for the combined dataset using the Nutrient + Text database matching is quite successful and is an improvement over the status quo of manual lookup or using food group or intake variables.

## Figures and Tables

**Figure 1 nutrients-11-03045-f001:**
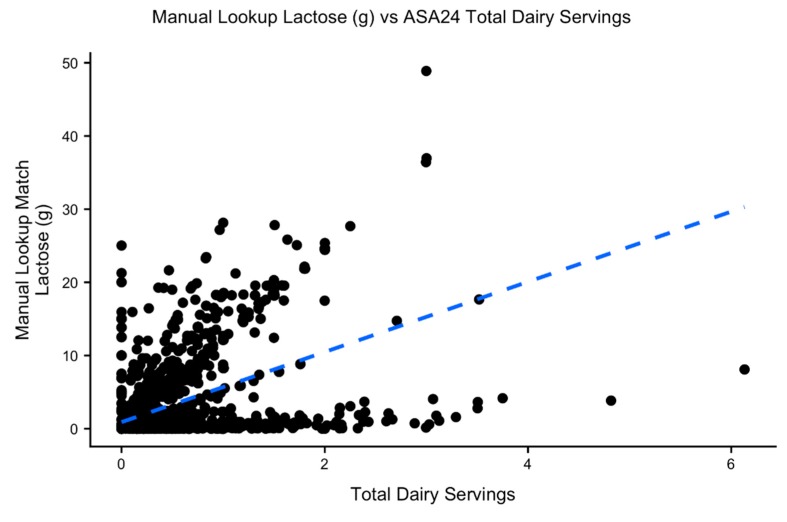
Comparison of estimated lactose to the servings of dairy for foods reported in the Automated Self-Administered 24-Hour Dietary Assessment Tool (ASA24) by a cohort of healthy U.S. adults. Lactose was estimated by manually looking up ASA24-reported foods into Nutrition Data System for Research (NDSR). The servings of total dairy are from the standard ASA24 output, with servings of soymilk subtracted [10,11]. The blue dashed line indicates the linear line of best fit.

**Figure 2 nutrients-11-03045-f002:**
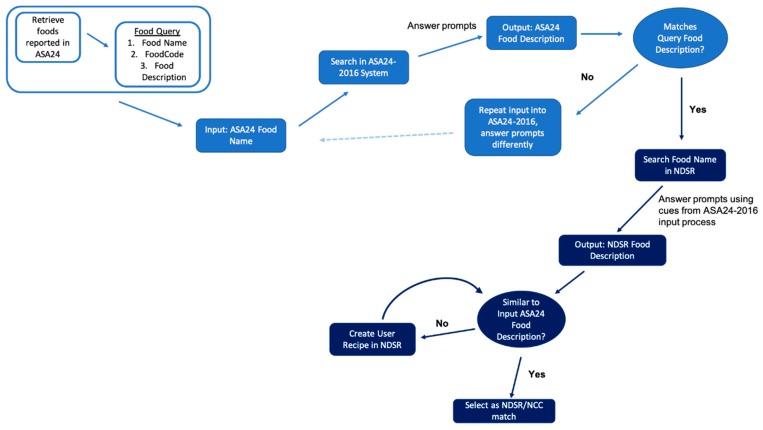
Overview of the manual lookup process. Query foods were selected from foods that were retrieved from an ASA24 output (white boxes). Each query food has three parts: (1) The Food Name, (2) the FoodCode, and (3) the corresponding Food Description. The manual lookup process can be broken down into two parts: lookup in ASA24 (light blue) and lookup in NDSR (dark blue). The ASA24 Food Name for each input food was searched in ASA24-2016 to retrieve the answers for each prompt to yield a given Food Description. The input Food Name was then searched in NDSR. The prompt/answer pairs obtained from ASA24-2016 inputs was used as a guide to selection answers from NDSR prompts. If the resulting NDSR output Food Description and the ASA24 Food Description were not similar, then a User Recipe was created in NDSR to serve as the match. NCC: Nutrition Coordinating Center.

**Figure 3 nutrients-11-03045-f003:**
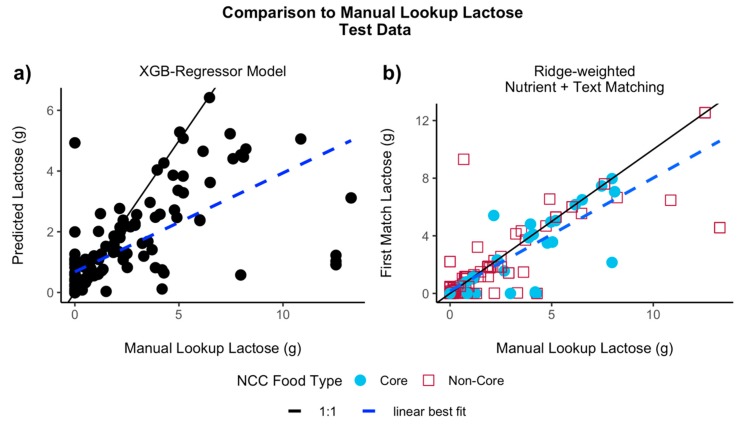
Comparison of the lactose (g) from the manual lookup to the (**a**) prediction from the XGB-Regressor model and (**b**) the Ridge-weighted Nutrient + Text database matching. For (**b**), markers are colored according to whether the first match was an NCC core food or a non-core food. Example of core foods are apples, honey, and bread. Examples of non-core foods are Genoa salami, cheese bread, and scrambled eggs.

**Figure 4 nutrients-11-03045-f004:**
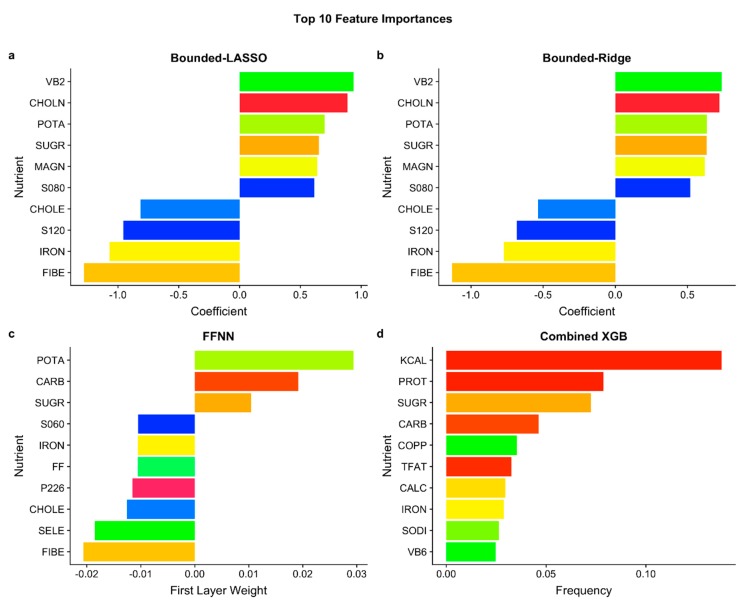
Top ten feature importances for the (**a**) Bounded-LASSO (least absolute shrinkage and selection operator), (**b**) Bounded-Ridge, (**c**) feed forward neural network (FFNN), and (**d**) Combined eXtreme Gradient Boosting (XGB) models. Feature importances were selected based on the absolute value of the coefficient/weight/frequency but the actual value is plotted. The frequency values for the XGB model are always positive. VB2: Vitamin B2; CHOLN: Choline; POTA: Potassium; SUGR: Sugar; MAGN: Magnesium; S080: Octanoic Acid; CHOLE: Cholesterol; S120: Dodecanoic Acid; FIBE: Fiber; FF: Folate; P226: Docosahexaenoic Acid (DHA); SELE: Selenium; PROT: Total Protein; COPP: Copper; CARB: Total Carbohydrate; TFAT: Total Fat; CALC: Calcium; SODI: Sodium; VB6: Vitamin B6.

**Table 1 nutrients-11-03045-t001:** Performance of the machine learning models.

Performance Metric	LASSO	Bounded-LASSO ^a^	Combined LASSO ^b^	Ridge	Bounded-Ridge ^a^	Combined Ridge ^b^	FFNN	XGB-Regressor	Combined XGB ^b^
**Training Results ***									
***R*** **^2^**	0.45	0.55	0.64	0.42	0.53	0.64	0.74	*0.81*	*0.81*
**SRC**	0.61	0.64	0.80	0.60	0.63	0.81	0.77	0.84	*0.90*
**PCC**	0.70	0.75	0.82	0.69	0.74	0.82	0.88	0.90	*0.91*
**MAE**	1.23	1.08	0.86	1.23	1.10	0.87	0.69	0.48	*0.43*
**Classifier Accuracy**	NA	NA	0.89	NA	NA	0.89	NA	NA	*0.97*
**Test Results**									
***R*** **^2^**	0.32 (0.36)	0.30 (0.53)	0.32 (0.54)	0.15 (0.27)	0.12 (0.50)	0.18 (0.46)	0.27 (0.28)	*0.33* (0.33)	0.31 (0.31)
**SRC**	0.64 (0.67)	0.63 (0.68)	0.71 (0.75)	0.61 (0.66)	0.64 (0.69)	0.72 (0.77)	0.70 (0.69)	0.75 (0.76)	*0.76* (0.76)
**PCC**	0.62 (0.67)	0.58 (0.78)	0.61 (0.80)	0.47 (0.59)	0.48 (0.76)	0.51 (0.77)	0.60 (0.60)	*0.65* (0.65)	0.64 (0.63)
**MAE**	1.53 (1.49)	1.37 (1.24)	1.28 (1.16)	1.64 (1.56)	1.43 (1.27)	1.37 (1.23)	1.3 (1.28)	1.18 (1.19)	*1.14* (1.16)
**Classifier Accuracy**	NA	NA	0.85 (0.86)	NA	NA	0.85 (0.86)	NA	NA	*0.92* (0.91)

Values in ( ) are results removal of “Salmon, raw” (ASA24 FoodCode 26137100). SRC: Spearman Rank Coefficient; PCC: Pearson’s Correlation Coefficient; MAE: Mean Absolute Error. Italicized values indicate the highest *R*^2^, SRC, PCC, and Classifier Accuracy, and the lowest MAE for the training or test data sets. ^a^ negative predictions were clipped to zero, ^b^ Combined models include a classifier and regressor, * Training results are the averages from 10-fold cross validation.

**Table 2 nutrients-11-03045-t002:** Comparison of the lactose (g) between the manual lookup and first match for each database matching algorithm.

Matching Algorithm	Weighting	Training (*n* = 378)				Test (*n* = 189)				All Data (*n* = 567)			
		*R* ^2^	PCC	MAE	Variation	*R* ^2^	PCC	MAE	Variation	*R* ^2^	PCC	MAE	Variation
Nutrient-Only	Unweighted	*0.51*	0.71	*0.71*	*0.39*	*0.32*	*0.58*	*1.18*	*0.52*	*0.44*	*0.66*	*0.87*	*0.44*
	LASSO-weighted	0.31 (0.47)	0.55 (0.69)	1.05 (0.93)	0.66 (0.68)	0.19	0.44	1.45	0.77	0.25 (0.36)	0.50 (0.60)	1.18 (1.11)	0.70 (0.70)
	Ridge-weighted	*0.51*	*0.72*	0.74	0.48	*0.32*	0.56	1.24	0.61	*0.44*	*0.66*	0.91	0.50
Nutrient + Text	Unweighted	*0.78*	*0.88*	*0.48*	0.60	0.72	0.85	0.69	*0.54*	*0.76*	*0.87*	*0.54*	0.58
	LASSO-weighted	0.20 (0.57)	0.44 (0.76)	0.92 (0.66)	0.60 (0.59)	0.64	0.80	0.78	0.63	0.26 (0.58)	0.51 (0.76)	0.90 (0.73)	0.60 (0.60)
	Ridge-weighted	0.23 (0.65)	0.48 (0.81)	0.89 (0.63)	*0.54* (0.54)	*0.75*	*0.86*	*0.62*	*0.54*	0.31 (0.68)	0.56 (0.82)	0.81 (0.64)	*0.54* (0.54)

PCC: Pearson’s Correlation Coefficient; MAE: Mean Absolute Error. The variation represents the median coefficient of variation in the g of lactose among the top five matches returned by each algorithm. Values in () are after the hot cocoa “outliers” are removed. Italicized values indicate the highest *R*^2^ and PCC and lowest MAE and variation values for the nutrient-only and nutrient + text algorithms for each dataset.

**Table 3 nutrients-11-03045-t003:** Comparisons of first matches among unweighted, LASSO-weighted, and Ridge-weighted. Nutrient-Only and Nutrient + Text matching.

Input ASA24 Food Description	NDSR/NCC Manual Lookup	Matching Scheme	First MatchNCC Short Description			g Lactose per 100g of Food			
			Unweighted	LASSO-Weighted	Ridge-Weighted	Manual Lookup	Un-Weighted	LASSO-Weighted	Ridge-Weighted
Milk, cow’s, fluid, whole	Milk, whole (3.5–4% fat)	Nutrient-Only	Milk, lactose reduced (Lactaid), whole	Milk, lactose reduced (Lactaid), whole	Milk, lactose reduced (Lactaid), whole	5.05	0.00	0.00	0.00
		Nutrient + Text	Milk, whole	Milk, whole	Milk, whole	5.05	5.05	5.05	5.05
Milk, cow’s, fluid, lactose reduced, 2% fat	Milk, lactose reduced (Lactaid), 2% fat or reduced fat	Nutrient-Only	Arby’s milk	Milk, acidophilus, 2% fat (reduced fat)	Milk, lactose reduced (Lactaid), 2% fat (reduced fat)	0.00	5.01	5.01	0.00
		Nutrient + Text	Milk, lactose reduced (Lactaid), 2% fat (reduced fat)	Milk, lactose reduced (Lactaid), 2% fat (reduced fat)	Milk, lactose reduced (Lactaid), 2% fat (reduced fat)	0.00	0.00	0.00	0.00
Milk, soy, ready-to-drink, not baby’s	Milk, soy milk, ready-to-drink, plain or original, unknown if sweetened, unknown sweetening, unknown type, unknown if fortified	Nutrient-Only	Soy milk, plain or original, sweetened with sugar, ready-to-drink, enriched	Soy milk, vanilla or other flavors, sweetened with sugar, ready-to-drink, enriched	Soy milk, vanilla or other flavors, sweetened with sugar, ready-to-drink, enriched	0.00	0.00	0.00	0.00
		Nutrient + Text	Soy milk, chocolate, sweetened with sugar, light, ready-to-drink	Soy milk, chocolate, sweetened with sugar, ready-to-drink, not fortified	Soy milk, chocolate, sweetened with sugar, ready-to-drink, not fortified	0.00	0.00	0.00	0.00
Milk, almond, ready-to-drink	Milk, almond beverage, plain or original, unknown type	Nutrient-Only	Cashew milk, chocolate	SnackWell’s 100 Calorie Pack—Fudge Drizzled Double Chocolate Chip (Nabisco)	Cashew milk, chocolate	0.00	0.00	0.00	0.00
		Nutrient + Text	Almond milk, chocolate, sweetened	Chocolate milk, ready-to-drink	Almond milk, chocolate, sweetened	0.00	0.00	5.12	0.00
Cheese, Cheddar	Cheddar cheese, unknown type	Nutrient-Only	Colby cheese, natural	Pepper Jack cheese	Colby Jack cheese	0.12	0.23	0.67	0.37
		Nutrient + Text	Cheddar cheese, natural	Cheddar cheese, natural	Cheddar cheese, natural	0.12	0.12	0.12	0.12
Yogurt, plain, whole milk	Yogurt, plain, whole milk (3%–4% fat)	Nutrient-Only	Mountain High Original Style Yoghurt—plain	Mountain High Original Style Yoghurt—plain	Stonyfield Organic YoBaby Yogurt—plain	3.38	3.38	3.38	3.38
		Nutrient + Text	Yogurt, plain, whole milk	Yogurt, plain, whole milk	Yogurt, plain, whole milk	3.38	3.38	3.38	3.38
High protein bar, candy-like, soy and milk base	Special formulated products, Tiger’s Milk Nutrition Bar—Protein Rich	Nutrient-Only	Tiger’s Milk Nutrition Bar—Peanut Butter and Honey	Tiger’s Milk Nutrition Bar—Peanut Butter and Honey	Tiger’s Milk Nutrition Bar—Peanut Butter and Honey	3.24	4.52	4.52	4.52
		Nutrient + Text	Nutribar High Protein Meal Replacement Bar—Milk Chocolate Peanut	Slim-Fast High Protein—Creamy Milk Chocolate, dry mix (unprepared)	High-protein bar	3.24	1.54	12.19	0.12
Clam chowder, NS as to Manhattan or New England style	Clams, soup—clam chowder, New England (cream base), unknown preparation	Nutrient-Only	Pescado frito con mojo (fish a la creole)	Dairy Queen Hot Dog with chili	Cheeseburger on a bun, single patty (1/10 LB), with ketchup, tomato, lettuce, pickle, onion, mustard	2.46	0.00	0.00	0.17
		Nutrient + Text	Manhattan clam chowder, tomato base, homemade	Manhattan clam chowder, tomato base, ready-to-serve can	Manhattan clam chowder, condensed	2.46	0.00	0.00	0.00
Cocoa, sugar, and dry milk mixture, water added	Cocoa or hot chocolate, prepared from dry mix, unknown type	Nutrient-Only	Soy milk, vanilla or other flavors, sweetened with sugar, light, ready-to-drink, not fortified	Almond milk, vanilla or other flavors, sweetened	Gerber Breakfast Buddies Hot Cereal with Real Fruit and Yogurt—Bananas and Cream	0.92	0.00	0.00	2.80
		Nutrient + Text	Land O’Lakes Cocoa Classics—Arctic White Cocoa, prepared	Swiss Miss Hot Cocoa Sensible Sweets—No Sugar Added, dry mix (unprepared)	Swiss Miss Hot Cocoa Sensible Sweets—No Sugar Added, dry mix (unprepared)	0.92	0.89	51.88	51.88

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
