# Peer review of "Nutrient Estimation from 24-Hour Food Recalls Using Machine Learning and Database Mapping: A Case Study with Lactose"

_nutrients, 2019, doi:10.3390/nu11123045_

Round 1

Reviewer 1 Report

Thank you for the opportunity to review this manuscript. Despite the technical nature of the study, the text is highly readable, aided by the quality and clarity of the figures and tables. Although I cannot comment on the appropriateness of the modifications made to the linear regression and the choice of learning algorithms, the statistical analyses, despite their complexity, are particularly clearly reported. I have read this paper carefully, and in my opinion it is mostly very well presented, however I have a small number of suggestions that may improve the overall clarity.

Introduction

Revise the use of the term nutrient when referring to dietary/food components more generally. “studying the link between diet and LP genetics is limited…” expand a little on why this area of research is important as context to your choice of lactose for your case study. ‘American cheese food’ – not clear - please amend or explain. Lines 78-86. This paragraph could be edited down to the most relevant 1-2 sentences. Lines 111-114. The novelty of the current study is explicitly stated here. However, the length of the sentence obscures the meaning – re-phrase into two sentences for clarity. The purpose of the study could more clearly be stated as an overall aim and a set of objectives.

Materials and methods

‘at least 0.1 cup equivalent..’ please translate to g/mls for non-US readers. The comparisons by which the machine learning and database matching are judged are stated but it would be useful if these ‘gold standard(s)’ were more clearly highlighted in one place.

Results

1 and 3.1.1: there is information here that reads as repetition of the methods. 2: as for item 9, but also appear to have discussion points here. In addition to the accuracy of the machine learning and matching methods, is it possible to also report the extent of the time-saving achieved in using these methods compared to manual look-up.

Discussion

A summary of the findings is useful here but repetition of the study context, aims, and methods is not necessary. Where integrating findings into previous research there is no need to repeat the descriptive detail of the studies reviewed in the introduction – the discussion points could therefore be much more concise. Figure 4 is misplaced here and should be in the results. Review the discussion overall and remove points that refer to the study limitations to avoid repetition in the designated Limitations section. Avoid negative contractions i.e. use ‘do not’ rather than ‘don’t’. Amend the conclusion so that it is a concise summary of the key points of the study, in particular removing reference to figures (and much of the second paragraph which is not needed).

Reviewer 2 Report

Dear Authors,

thank you for providing such an interesting paper on possible machine learning models to use in prediction of lactose content in food products, as a representative of any other nutrient.

The paper is well organised, with a comprehensive Introduction on available nutrient estimation tools; a detailed section "Materials and Methods" where the reader gets acquainted with the steps you planned and you aim to inquire; and finally an extensive description of outcomes. The paragraph on limitations of your work and future perspectives usefully complements the manuscript.

Please, add in line 93 and 96 two references.
